# Development of a Dental Implantable Temperature Sensor for Real-Time Diagnosis of Infectious Disease

**DOI:** 10.3390/s20143953

**Published:** 2020-07-16

**Authors:** Jeffrey J. Kim, Gery R. Stafford, Carlos Beauchamp, Shin Ae Kim

**Affiliations:** 1American Dental Association Science & Research Institute, American Dental Association, Gaithersburg, MD 20899, USA; kimj@ada.org; 2Materials Science and Engineering Laboratory, National Institute of Standards and Technology, Gaithersburg, MD 20899, USA; gery.stafford@nist.gov (G.R.S.); carlos.beauchamp@nist.gov (C.B.)

**Keywords:** implantable temperature sensor, dental implants, peri-implantitis, flexible sensor, polyimide film, microfabrication, local and continuous monitoring, early diagnosis

## Abstract

Implantable sensors capable of real-time measurements are powerful tools to diagnose disease and maintain health by providing continuous or regular biometric monitoring. In this paper, we present a dental implantable temperature sensor that can send early warning signals in real time before the implant fails. Using a microfabrication process on a flexible polyimide film, we successfully fabricated a multi-channel temperature sensor that can be wrapped around a dental implant abutment wing. In addition, the feasibility, durability, and implantability of the sensor were investigated. First, high linearity and repeatability between electrical resistance and temperature confirmed the feasibility of the sensor with a temperature coefficient of resistance (TCR) value of 3.33 × 10^–3^/°C between 20 and 100 °C. Second, constant TCR values and robust optical images without damage validated sufficient thermal, chemical, and mechanical durability in the sensor’s performance and structures. Lastly, the elastic response of the sensor’s flexible substrate film to thermal and humidity variations, simulating in the oral environment, suggested its successful long-term implantability. Based on these findings, we have successfully developed a polymer-based flexible temperature sensor for dental implant systems.

## 1. Introduction

Dental implants are becoming preferred restorative options to replace missing teeth. Despite its high success rate, patients with dental implants experience infectious diseases such as peri-implantitis [1,2,3,4]. Peri-implantitis involves an inflammatory process around a dental implant, which includes both soft tissue inflammation and progressive loss of supporting bone beyond biological bone remodeling [2]. If not diagnosed early and treated in a timely manner, the inflammation may lead to implant failure, which requires a second implant surgery [3]. 

Current diagnostic methods for peri-implantitis are based on the signs and symptoms of the disease. In fact, clinicians investigate color change and/or bleeding in the gingiva, measure probe depth of implant pockets, and review radiographs to determine the bone loss. However, these methods can be subjective, lack precision, and are time-delayed until the patient already has symptoms and visits a dental clinic. Moreover, there is no consensus on which set of clinical parameters are most important in diagnosing peri-implantitis accurately and predicting the success of treatment [4,5,6]. 

To obtain useful clinical data in a timely manner, a point-of-care (POC) system with real-time and off-site diagnosis is of great interest [7,8]. For example, glucose sensors for diabetic monitoring and lateral flow sensors for pregnancy tests are widely used POC systems. With POC platforms, the time required for diagnosis would be shorter than visiting a dental clinic after experiencing symptoms, and the saved time could facilitate the clinical decision-making process. Moreover, a POC system requires no expertise in sample analysis. Despite these benefits, it is challenging to perform continuous or regular biometric monitoring with conventional POC platforms due to technical barriers, especially size and sensing mechanism. 

One successful solution to overcome this technical barrier is to integrate nano-/micro- technologies into the POC devices. Various types of wearable or implantable devices have been developed successfully that allow real-time and continuous health monitoring for at home self-care management. For in vivo oral health monitoring, sensors have been integrated into denture, orthodontic braces, and mouth-guard platforms to detect pH [9], humidity [10], specific chemicals [11,12], and salivary metabolites [13,14]. However, their sensitivity was limited by sampling saliva from the entire mouth and lacked local detection targeting a particular tooth or implant of clinical interest. A peptide-coated graphene nanosensor was another diagnostic platform that operated by attachment on the tooth enamel surface for bacterial monitoring in saliva [15]. However, this approach required immobilization of peptides, which may increase fabrication cost and could allow non-specific binding that may lead to false readings. 

In addition to chemical and biological changes, temperature is a targeting parameter that correlates with inflammation. In fact, various types of temperature sensors and fully implantable or wearable supporting systems have been developed and successfully applied for monitoring in various organs such as cardiac tissue [16], abdomen [17], intracranial surface [18], and skin [19,20]. For oral diagnostics, a periodontal temperature probe has been developed for detecting small temperature changes on localized periodontal sites [21]. However, the temperature probe was designed to be used in dental offices to measure temperature at a given time. It is not possible to continuously monitor temperature in the oral cavity. Another research group developed an intraoral appliance which is capable of measuring the temperature continuously over 24 h using a general-purpose thermocouple sensor [22]. However, this appliance is limited by having external and bulky wire connections that made wearers uncomfortable. A poly(N-isopropylacrylamide)-based hydrogel was used to develop a tiny tooth-mounted sensor that could calculate temperature change based on its interlayer volume change [23]. However, the interlayer material of the sensor was responsive to various type of analytes, pH, and temperature, and thus resulted in a lack of linearity and specificity. Therefore, despite prior efforts, an implantable temperature oral sensor which is user-friendly, sensitive and stable is still needed for local and continuous or regular biometric monitoring around dental implants.

To address these requirements, we developed a micro-temperature sensor for a dental implant application that could provide local and regular monitoring in the daily life of wearers. We have optimized the design and microfabrication process of the sensor that can be produced in a timely and cost-effective manner. Lastly, we tested the performance and stability of the sensor.

## 2. Materials and Methods

### 2.1. Sensor Materials and Design

Many types of commercially available polyimides have been widely used for biosensor encapsulation and as substrates for implantable devices due to their high flexibility, electrical resistivity, thermal and chemical stability, and biocompatibility [24,25,26]. In addition to these benefits, photo-definable polyimides are easy to pattern using photo-lithography, which facilitates biosensor or bioelectronics fabrication [25,26,27,28]. Therefore, in this study, photo-definable polyimide (HD-8820, HD MicroSystems, Parlin, NJ, USA) served as both structural substrates and insulation layers that resulted in simplified fabrication and cost savings. To note, we have tested the biocompatibility of the polyimide in primary mouse epithelial tongue cell growth, but no cytotoxicity was found (Appendix A).

The schematic design of the implantable temperature sensor using polyimide is shown in Figure 1. As shown in Figure 1a, the temperature sensor is designed as a multi-channel mechanism to measure the localized temperature change on four different points, namely, buccal, lingual, mesial, and distal points, of a dental implant. Since the sensor adheres around the abutment wing of the dental implant where inflammation is likely to occur, the horizontal length and width of the sensor was 15 and 1 mm, respectively. These dimensions and the shape of the sensor, as well as the number of channels, can easily be modified depending on the size of the abutment wing and dental implant. Each channel has four gold pads to contact with a four-wire measurement circuit. For fully real-time diagnosis, the pads connect to active components under the implanted crown with full insulation. Figure 1b is a design of the platinum (Pt) resistor and the gold (Au) interconnection lines. An additional layer of chromium (Cr) provides adhesion between the Pt or Au noble metals and the polyimide film. Pt was selected as the sensing material due to its high linear correlation between electrical resistance and temperature. Au was chosen as the conductive material and contact pad due to its biocompatibility along with electrical and chemical stability. To minimize self-heating and power consumption during the temperature sensing, the maximum Pt resistance was achieved by a long serpentine design (20 vertical lines) with 3 μm width and 6 μm pitch. To increase the current density uniformity on the serpentine design, the corner was also modified as a round edge, as shown in the inset of Figure 1b. Forty-six sensors are placed together on a single 100 mm diameter wafer, as shown in Figure 1c. To minimize sensor damage during final release from the wafer, a 2 mm width vertical bar-shaped structure was designed to provide support to the sensors with a 200 μm width small notch connecting each sensor to the structure. The notch width was optimized so that it is rigid enough to support each sensor, but small enough to snap during individual detachment from the supporting structure.

### 2.2. The Microfabrication Process

A schematic diagram for the microfabrication process is presented in Figure 2. Fabrication starts with silicon oxide deposition on a standard 100 mm round silicon wafer (Figure 2a). A 1 μm thick silicon oxide was deposited using plasma-enhanced chemical vapor deposition (Versaline PECVD, Plasma-Therm, Petersburg, FL, USA). The silicon oxide serves as a sacrificial layer to release the sensors from the silicon substrate, which serves as a mechanical support during the entire fabrication process. A 5 μm thick polyimide film was coated on top of the sacrificial layer. The photo-definable polyimide was spin coated at a speed of 314 rad/s for 60 s, followed by exposure at 350 mJ/cm^2^ dose using g-line contact aligner (MA6/BA6, Suss MicroTec, Garching, Germany) to define outline geometry of the sensors. The polyimide film was then cured at 275 °C for 2 h under a nitrogen atmosphere in a nitrogen oven (Heratherm, Thermo Fisher Scientific, Waltham, MA, USA) (Figure 2b). To reduce thermal stress in the polyimide film, temperature ramping and cooling rates were controlled at approximately 4 °C/min using a programmable oven, followed by room temperature stabilization for 24 h. The Cr/Pt (20 nm/200 nm) temperature sensing resistor was patterned using a lift-off process. Positive photoresist (SPR220-3.0™, Dow, Midland, TX, USA) was first patterned on top of the polyimide film, followed by Cr/Pt deposition using an e-beam evaporator (Infinity 22, Denton Vacuum, Moorestown, NJ, USA). The wafer was then soaked in acetone and isopropanol for a few minutes to strip the photoresist left behind the patterned Cr/Pt (Figure 2c). The same process was applied to the Cr/Au (20 nm/250 nm) patterning for conductive interconnection lines and pads (Figure 2d). After metallization, a top insulating polyimide layer was spin-coated, patterned, and baked under the same conditions as the first polyimide layer (Figure 2e). Since the metal contact pads were exposed during the patterning process of photo-definable polyimide, an additional opening process with dry etching was not required. Lastly, silicon oxide was wet etched in buffered oxide etchant (BOE 6:1) to release the sensors from the silicon wafer, as shown in Figure 2f. After rinsing and drying the released sensors, the surface and metal patterns were examined under an optical microscope (L200, Nikon, Tokyo, Japan) and a scanning electron microscope (SEM JSM-IT 500, JEOL, Tokyo, Japan). 

In summary, our fabrication process of the dental implantable temperature sensors used photo-definable polyimide with the optimized lift-off process that eliminated four steps of dry etching and mask patterning processes resulting in 56% reduction in fabrication cost. 

### 2.3. Temperature Measurement Circuit and Setup

To increase accuracy in the temperature measurement without contribution from the Au lines, a four-wire measurement circuit was employed, as shown in Figure 3a. In the four-wire configuration, the current is passed through the outer lines (L1 and L4) and the voltage drop is measured across the inner lines (L2 and L3). Therefore, the Au resistance has minimal effects on the voltage measurement which correlates resistance to the temperature. To facilitate contact between the sensor and the circuit, a printed circuit board (PCB) that included a 16-pin flexible flat cable connector was designed and assembled by the PCB manufacturer. Through the PCB, a constant current of 1 mA was applied with a current source (Keithley 6221, Keithley Instruments, Cleveland, OH, USA), and the voltage deviation was monitored by a digital multimeter (2480R, Data Precision, Wakefield, MA, USA) in a temperature-controlled small chamber filled with deionized (DI) water or other solutions. To note, the current will be ultimately reduced lower than 0.5 mA to meet medical grade standard and save power. Temperature was controlled from 20 to 100 °C, with 1 °C precision (Super-Nuova Stirring Hotplate, Barnstead Thermolyne, Ramsey, NJ, USA). To minimize the temperature variation between the temperature controller and DI water, the temperature was simultaneously monitored with a commercial thermometer (52K/J, Fluke Corporation, Everett, WA, USA) in the bath positioned within 1 mm of the fabricated sensor. 

### 2.4. In Situ Cantilever Curvature Measurement System

To measure the thermal and relative humidity (RH) stress response of the polyimide film, in situ cantilever curvature was used. To preapare a polyimide-coated cantilever that replicates our sensor substrate, a polyimide film was spin-coated onto a 200 μm thick double-side polished silicon wafer (100) and cured as described in Figure 2b. After 24 h stabilizing time at room temperature, the cantilevers were cut into lengths and widths of 6 cm and 4 mm, respectively. These silicon cantilevers had a Young’s modulus of 130 × 10^9^ N/m^2^ and Poisson’s ratio of 0.28. The thickness of the polyimide film was measured by SEM imaging and by profilometry (Dektak XT, Bruker, Billerica, MA, USA). Since the metal patterning process was excluded from polyimide fabrication on the cantilever, there was a difference of 0.2 μm compared to the thickness of an actual sensor.

The polyimide-coated silicon cantilever was vertically mounted in a 25 mL Pyrex chamber that allowed for controlled air flow. The chamber was mounted and aligned on an optical bench. For thermal control inside the chamber, heating was applied by a computer-controlled resistive heater located in the chamber while cooling was accomplished naturally under a constant flow of dry air. For RH control inside the chamber, two mass flow controllers (MKS type 146C, MKS Instruments, Andover, MA, USA) in parallel were used to mix dry (0% RH) and humid air. To form humid air, dry air from a gas cylinder was bubbled through a DI water reservoir. The typical RH range was approximately 2% to 80% [29,30]. The temperature and RH were measured using a digital termperature and humidity sensor (SHT85, Sensirion AG, Stäfa, Switzerland), placed in the chamber and positioned approximately 2 mm from the cantilever. All experiments were computer-controlled using in-house LabView software. 

Cantilever curvature was monitored using a multi-beam optical stress sensor (MOSS), similar to that described in previous studies [31]. A schematic is shown in Figure 3b. The apparatus consists of an 18 mW AlGaInP (658 nm) diode laser, an etalon that generates three parallel laser beams, a beam splitter, mirrors, servos for mirror control, and a CCD video camera. The cantilever curvature, κ, is given by
(1)$$\kappa =\frac{\Delta D}{D_{o}}\frac{\cos\left(\alpha \right)}{2L}$$
where ΔD/D_o_ is the change in the average spacing between adjacent laser spots reflected onto the CCD camera, normalized by the initial spacing, L is the distance between the cantilever and CCD, and α is the angle of incidence (typically equal to zero). Two computer-controlled servos (Kinesis, Thorlabs, Newton, MA, USA) center the laser spots onto the CCD to negate thermal drift. The typical curvature resolution is 2 × 10^–4^ m^–1^ (5 km radius of curvature). If D_o_ is chosen to be the spot separation produced by a flat mirror (κ = 0) rather than an arbitrary initial value, then the actual κ of the cantilever can be measured, as can the actual stress in the film.

The relationship between the force per cantilever beam width (ΔF_c_) exerted by the polyimide film and the radius of curvature of the cantilever is given by Stoney’s formula [32], which assumes that the polyimide film is considerably thinner than the Si substrate,
(2)$$\Delta F_{c}=\frac{M_{s}h_{s}^{2}\kappa }{6}=\sigma h_{f}=M_{f}\varepsilon h_{f}$$
where M_s_(M_f_) and h_s_(h_f_) are the biaxial elastic modulus and thickness of the substrate(film), σ is the average biaxial stress of the polyimide film and ε is the elastic strain in the film. The thin-film approximation introduces an error of 2.3%. Since, the derived cantilever force (ΔF_c_) is equal to the average biaxial stress of the polyimide film (σ) multiplied by its thickness (h_f_), ΔFc will be referred to as the stress-thickness. The 2 × 10^–4^ m^–1^ resolution in κ mentioned above corresponds to a stress-thickness resolution of approximately 0.06 N/m. 

### 2.5. Reagents

To test the chemical stability of the sensors, six solutions were prepared: DI water, lab-made artificial saliva (SAGF recipe [33]: 0.13 g/L NaCl, 0.96 g/L KCl, 0.66 g/L KH_2_PO_4_, 0.63 g/L NaHCO_3_, 0.19 g/L KSCN, 0.23 g/L CaCl_2_·2H_2_O, 0.20 g/L Urea, and 0.76 g/L Na_2_SO_4_·10H_2_O), mouth wash (Listerine), soda (Coca Cola), orange juice (Tropicana 100%), and brewed black coffee. The artificial saliva also contains a low concentration of sodium azide (<1.12 mg/ml) to inhibit bacterial growth [34]. To mimic oral temperature, sensors were immersed and tested at 37 °C in an incubator for 6–7 d. The TCR values and microscopic images of each sensor were compared before and after chemical exposure. 

## 3. Results

### 3.1. Microfabricated Temperature Sensors

Using the microfabrication procedure, 46 sensors were successfully fabricated from a 100 mm silicon wafer, as shown in Figure 4a. After releasing the sensors from the wafer, each sensor was connected to a lab-designed PCB and tested for resistance. Based on the measurements from multiple batches, the yield exceeded 95%, and the measured resistance (655.5 ± 11.7) Ω (mean ± SD) was close to the designed (618.0 Ω) value, based on the length, cross-section area, and resistivity of Pt, at room temperature (20 °C) with high repeatability. The difference (37.5 Ω) between the measured and the designed resistance may originate from variations in deposition rates of the e-beam evaporator which affect the cross-sectional area of the Pt pattern. In contrast to the possibility of variations in cross-sectional area, the width of the Pt pattern was uniform (3 μm), as shown in Figure 4b. In addition, each sensor was small enough for full integration on dental implants with high flexibility due to the thin polyimide film (Figure 4c–e). This optimized sensor design and high flexibility allowed easy adherence around the abutment wing (Figure 4d) without physical damage to either metal lines or polymer film. Based on the sensor size and flexible substrate properties, the sensor is implantable on dental implants and compatible for localized monitoring of intraoral temperature changes.

### 3.2. Temperature Sensing Analysis

To analyze the temperature sensing capability of the microfabricated sensors, a correlation between the surrounding temperature change in DI water and its consequent resistance change was tested among randomly selected 20 sensors (Figure 5a). The four-wire circuit (Figure 3a) was used to measure the resistance change of the sensor while the temperature range changed from 20 to 100 °C. The temperature range fully covered the target operating range for dental implant applications (20 to 50 °C) and was sufficient to identify any reliability and stability failure modes at a higher temperature range. When plotting all the data from 20 sensors, the average Pt resistance was linearly correlated to the temperature, as shown in Figure 5a. In addition, although the resistance of each sensor varied at all temperatures, the linearity in resistance response was universal (Figure 5b). The accuracy of the measured value of each sensor is ±0.2 °C at 20 °C during five repeated measurements and the accuracy is consistent with our target value to detect inflammation around the implant. Furthermore, the resistance change to temperature was independent of pH levels and the type of solution such as DI water, PBS, artificial saliva, because the Pt resistor does not chemically react with the solution (data not shown). Although the maximum operating temperature (≈ 200 °C) was limited by the polymer film, we expect the sensor to be sufficient for monitoring the intraoral environment (20 to 50 °C) including cold or hot beverages and food [22,35]. 

The temperature coefficient of resistance (TCR) was calculated between the measured temperature range (20 to 100 °C) using the following equation:(3)$$TCR=\frac{R_{2}-R_{1}}{R_{1}\times \left(T_{2}-T_{1}\right)}=\frac{R_{100℃}-R_{20℃}}{R_{20℃}\times \left(100-20\right)℃}\;\left({℃}^{-1}\right)$$

Based on the equation above, four randomly selected sensors (Figure 5b) showed similar TCR values of 3.43 × 10^−3^/°C (Sensor 1), 3.46 × 10^−3^/°C (Sensor 2), 3.44 × 10^−3^/°C (Sensor 3), and 3.23 × 10^−3^/°C (Sensor 4). The TCR value for each sensor was constant over repeated temperature measurements. Likewise, measurements from 20 sensors yielded an average TCR value of 3.33 × 10^−3^/°C, which is close to the bulk Pt TCR (3.85 × 10^−3^/°C). The small difference (0.52 × 10^−3^/°C) between the measured average and bulk TCR is speculated to be derived from additional Cr layers and minor contribution of Au not completely removed using the four-wire measurement circuit. Lastly, the TCR variance of 46 sensors from a single batch was below 2%. Based on the high linearity and constant TCR values, we conclude that the microfabricated sensors, with ±0.2 °C accuracy, are suitable for precise temperature monitoring for detecting inflammation, which is expected to increase gradually by 1.5 to 4 °C and remain for several days [20,36,37,38].

### 3.3. The Funtional Stability of Sensors

#### 3.3.1. Thermal Stability

To test the thermal stability of the microfabricated sensors, the thermal stress-derived resistance change was measured. The thermal stress was induced by a temperature increase from 20 to 100 °C for 100 subsequent repeats with a rapid ramping rate of ≈ 130 °C/min. This was achieved by directly contacting a sensor to the heater. The resistance change data from each 100 repeats were plotted to check any deviation. As shown in Figure 6, the resistance change of the sensor in the 1st, 20th, 50th and 100th repeats overlapped. In fact, the resistance change after the 100th repeat only deviated by 0.3% from that of 1st temperature sweep. Moreover, all 100 resistance change measurements showed a linear correlation to the temperature with a stable TCR value of (3.23 ± 0.01) × 10^–3^/°C. When we imaged the sensor after the 100th repeat using both bright-field microscope and SEM, no visible degradation or damage was observed on either the polymer film or the metal patterns. These results allow us to conclude that the sensor can stably and repeatably monitor the dynamic intraoral temperature environment. 

#### 3.3.2. Chemical Stability

Since certain food and beverages can potentially react with implanted sensors in the oral cavity, the chemical stability of a sensor is also critical. To test the chemical stability of the microfabricated sensors, we compared TCR measurements and microscopic images of the sensors before and after exposing them to a number of solutions. For chemical exposure, we immersed the sensors into DI water (pH 7.0), artificial saliva (pH 6.8), mouth wash (pH 4.3), soda (pH 2.6), orange juice (pH 3.8), and black coffee (pH 5.1) in a 37 °C incubator for a maximum of 7 d. Except for DI water and saliva, the chemicals were acidic and some (soda and orange juice) could lead to teeth demineralization. The incubation time of 7 d (168 h) is approximately equivalent to the time of gargling mouthwash for 30 s twice a day for 27.6 years.

TCR values were calculated and compared before and after immersion, as shown in Figure 7a. Impressively, the sensors performed similarly in all solutions. Although some variation existed, no statistical difference was observed in TCR values between before and after chemical exposure in all solutions. The small increased TCR in DI water, mouthwash, and orange juice after chemical exposure may have resulted from leftover residue on pad metals or measurement error instead of the effects of chemicals or the acidity. Although soda has the lowest pH of 2.6, the variation of TCR value was similar to that of artificial saliva and black coffee. 

To verify the effect of pH on the polyimide film, the sensor surface was optically examined before and after the chemical exposure. According to microscopic images (Figure 7b), the sensor in soda, the lowest pH solution, showed no visible difference before and after chemical exposure. Other chemicals also showed no deformation or stress-associated changes on both polyimide film and metal patterns. Furthermore, no functional or physical degradation has been observed after immersion in DI water or artificial saliva for over two months. Based on these data, the sensor is chemically stable in various solutions that it may be exposed to in the mouth.

#### 3.3.3. Mechanical Stability

Since the developed temperature sensor has a 10 μm thick polymer film that supports the 220 nm thick metal lines of the Pt resistor, mechanical stability was tested by applying external physical forces by bending and stirring motions. For applying external forces with a bending motion, the sensor was exposed to 180° bending by hand (Appendix A) and was repeated up to 1000 times. During the bending motion, it was evident that force was applied to the Pt resistors as well as the polyimide surface. However, the TCR value, as shown in Figure 8a, was consistent and showed no statistical differences between before and after the 1000 bending events. 

After implantation, the sensor may not be exposed to many mechanical stresses like that of the crown surface because the sensor will be statically adhered to the abutment wing of the dental implant platform and covered by soft tissue. However, chewing, flossing, and tooth brushing could potentially introduce mechanical impact and pressure on the implanted sensor. Therefore, to simulate the random mechanical impact and pressure with fast fluidic flow, stirring tests were performed. The sensor was fixed onto a magnetic bar and stirred at 31.4 rad/s for 30 min in DI water, as shown in Appendix A. After the stirring events, the TCR value was compared with the initial condition, as shown in Figure 8b. Similar to the bending test results, there was no statistical difference between the before and after stirring event in the TCR value. Furthermore, after both mechanical stress tests, neither surface damage nor disconnected metal lines were observed on microscopic images as well as Pt resistance measurements. Based on these results, the sensor is mechanically robust and durable.

### 3.4. The Physical Stability of Polyimide Film

#### 3.4.1. Thermal Stress Measurement 

In addition to the functional stability of the termperature sensor, the long-term mechanical stability was examined by measuring the thermal and humidity-induced response of the polyimide film using in situ cantilever curvature (Figure 3b). Since the polyimide film is rigidly fixed to the Si cantilever, in-plane expansion due to a temperature increase or water uptake is constrained, giving rise to a biaxial stress in the film. The stress can be quantified from the cantilever curvature through the Stoney equation (Equation (2)). Cyclic thermal stress was induced over a temperature range of 35 to 65 °C.

Chamber temperature (blue) and RH (red) over 22 thermal cycles, with each cycle duration being 40 min, is plotted in Figure 9a. The RH dropped below 5% under a flow of dry air before initiating the first heat cycle. For the duration of the thermal stress experiment, the RH remained in the 2% to 3% range. A plot of the corresponding stress-thickness response to the temperature change is shown in Figure 9b. The stress-thickness at the beginning of the experiment was arbitrarily set to zero. As the humidity dropped due to the flow of dry air through the chamber, the stress-thickness moved in the tensile direction to a maximum value of approximately 13 N/m. When considering the actual curvature of the cantilever and the 5.2 μm thickness of the film, this value corresponds to an initial polyimide biaxial film stress of 28.7 MPa. This value is very close to 37 MPa, as reported by the manufacturer's technical information when cured at 320 °C for 1 h, which is relatively higher in temperature and shorter in duration than the 275 °C for 2 h cure used in this study.

As the temperature increased, the stress-thickness moved in the compressive direction in Figure 9b. Compressive stress is expected to develop in the polyimide during heating due to its higher thermal expansion coefficient (59 × 10^−6^ K^−1^) with respect to that of silicon (2.6 × 10^−6^ K^−1^). Since the polyimide was rigidly held in place by the silicon cantilever and was unable to expand, compressive stress develops. The compressive stress was relieved as the temperature decreased during the natural cooling cycle. As the temperature cycles continued, the stress response was replicated for each cycle. This is a clear indication that within this temperature range, the polyimide film is elastic. Over the temperature range of 35 to 65 °C, the average stress-thickness change was approximately (–22 ± 0.4) N/m, corresponding to a biaxial stress change of approximately (−4.2 ± 0.07) MPa. Such a small stress change is clearly in the elastic region.

The thermal stress coefficient d(σh)/dT can be determined in two ways. The first involves taking the average Δ(σh)/ ΔT for each of the cycles. From the data shown in Figure 9a,b, the average value of Δ(σh)/ΔT was (−0.68 ± 0.02) N/(m°C). Alternatively, the stress-thickness change during a cooling cycle can be plotted, as shown in Figure 9c. The response was linear over the entire temperature range, with a slope of −0.70 N/(m°C), very similar to the averaged stress response. 

Since the elastic strain that develops in the polyimide film is given by ε = (α_s_ – α_f_) ΔT, where α_s_ and α_f_ are the thermal expansion coefficients for the silicon substrate and polyimide film, respectively, and ΔT is the temperature change, an expression for the thermal stress coefficient can be derived by inserting this strain into Equation (2) to yield,
(4)$$\frac{\sigma h_{f}}{\Delta T}=M_{f}h_{f}\left(\alpha_{s}-\alpha_{f}\right)$$

Assuming a polyimide biaxial modulus of 3.3 GPa and the thermal expansion coefficients listed above for polyimide and silicon, the thermal stress coefficient is −0.92 N/(m°C), which is slightly larger than the experimental value. Considering that the reported polyimide properties are very much dependent on the processing and curing conditions, some discrepancy between the measured and calculated values can be expected. 

#### 3.4.2. Humidity Stress Measurement

Since the oral environment has humidity variation as well as temperature, humidity stress on the polyimide film was also examined using the in situ cantilever curvature measurement system. Similar to the previous thermal stress analysis, the stress-thickness and humidity stress coefficients were measured while RH was varied between 2% and 80% inside the chamber by regulating the relative flow of dry and humidified air while maintaining a total flow rate of (500 ± 5) mL/min. Figure 10a shows a typical humidity profile obtained at room temperature. The accessible humidity range was dramatically reduced at elevated chamber temperatures due to laser scattering by condensation on the chamber walls. As such, we restrict this humidity stress evaluation to room temperature. The stress-thickness response to the controlled variation in RH shown in Figure 10a,b indicates that the polyimide film is quite sensitive to the RH. A plot of stress-thickness as a function of humidity is shown in Figure 10c. The data show a linear correlation with a humidity stress coefficient d(σh)/d(RH%) of −0.56 N/(m%). As expected, exposure to humidity produced compressive stress in the film. 

The overall stress-thickness change of 38.5 N/m is a clear indication that moisture is entering the bulk polymer. Although it is possible that water adsorbed onto the polymer surface could induce curvature, surface stresses are typically limited to a couple of N/m, far less than the stress change observed here. Assuming the moisture has access to the entire film, this stress-thickness change corresponds to a biaxial stress of −7.4 MPa. This relatively low stress change and the reversible response indicates that the polyimide is responding elastically to the humidity change. If the polyimide film is assumed to be isotropic, then the strain associated with volume expansion due to water uptake is given by
(5)$$\varepsilon ={\left(\frac{V}{V_{o}}\right)}^{\frac{1}{3}}-1$$
where *V_o_* is the volume of the dry polyimide film. Making use of Equation (2), the elastic strain can be calculated as 0.0022, which corresponds to a water uptake of 0.47% mass fraction (0.67% volume fraction). The moisture uptake for the polyimide is reported to be less than 0.5% for films cured at 320 °C for 1 h.

Based on these analyses, the polyimide film responds elastically to thermal and humidity stress expected to be encountered in the oral environment, consistent with the requirements of long-term oral implantation with high stability.

## 4. Discussion

In this study, we have successfully developed a dental implant temperature sensor with a microfabrication process that is capable of long-term monitoring in the oral environment. Our data show that the micro-scale implantable temperature sensor can stably conduct real-time measurements of temperature changes at the site of dental implants to send warning signals when inflammation occurs. In addition to the sufficient performance and high stability and repeatability, specific benefits of the sensor are as follows:Efficient localized monitoring. Compared to other saliva sensors, our sensor is small but efficient for onsite sensing with multi-channels around dental implants. The sensor can retrieve localized information from proximity soft tissue, where the inflammation is likely to occur.Flexible sensor design. By changing photomasks, additional channels can be easily integrated on the sensor without any outline change. Furthermore, other electrochemical or biological sensors could be integrated with the temperature sensor for various parameter sensing such as pH, bacteria, or specific biomarkers for advanced diagnosis.No calibration requirement. Unlike most lab-made sensors, our temperature sensor does not require calibration before measurement because it monitors the baseline change of temperature for a long period of time and is based on a reproducible resistance–temperature relationship. This strategy is beneficial because intraoral temperature is patient and location specific, and transient temperature changes due to events like drinking hot or cold beverages are not significant for disease diagnosis. Therefore, our sensor is effective for long-term monitoring of relative temperature changes.Safe for long-term implantation. Our sensor is built with previously tested biocompatible materials and shows a constant TCR value throughout testing without polymer film damage. In addition, the sensor pattern shows no deformation after exposure to both DI water and artificial saliva for two months at 37 °C. Furthermore, preliminary data using *Streptococcus mutans* (strain UA159) shows that biofilm formation on the sensor induced negligible temperature difference.

One additional consideration for a temperature sensor that was not discussed in this study is system integration including wireless data transfer and power delivery. In fact, system integration is one of the most actively studied areas for clinical applications such as glucose monitoring [39], oral activity recognition [40], and accelerating bone formation with electrical stimulation [41]. In such studies, the whole system integration was verified into a general dental implant platform for data transfer with a small battery for power delivery. However, rather large power delivery is necessary to facilitate wireless communication in implantable devices [10,42,43,44,45]. Future studies include wireless power delivery and battery-recharging capabilities that will reduce the risk of infection for battery replacement as well as the size of the integrated system. Furthermore, a current level and safety circuit of the integrated system will be refined to meet the medical-grade electrical safety standards. 

Therefore, we plan to further upgrade the sensor by further developing the microfabrication process and testing implantable capacity. First, we plan to integrate wireless communication and battery-recharging capabilities to the dental implant platform. Further, with further microfabrication development, we plan to add a physio-chemical sensing mechanism to the temperature sensor for simultaneous monitoring of multiple parameters. Lastly, we anticipate upgrading our sensor to a sensor-embedded chair-side dental instrument that can then be tested for oral disease diagnosis.

## 5. Conclusions

In this study, an optimized dental temperature sensor was developed from microfabrication using flexible polyimide film and noble metals. Its high linearity, repeatability, accuracy, and stability were verified by resistance measurement tests, microscopic observations, and in situ stress studies. Our data demonstrated that the sensor is stable in multiple stress conditions induced by dynamic temperature, acids, bending, and stirring. Furthermore, the polymer film behaved as linearly reversible during cyclic thermal- and humidity-induced stress. Based on these analyses, we have developed a successful oral implantable sensor and further believe that it can be used as a platform for future functional integration to provide customized diagnosis.

## Figures and Tables

**Figure 1 sensors-20-03953-f001:**
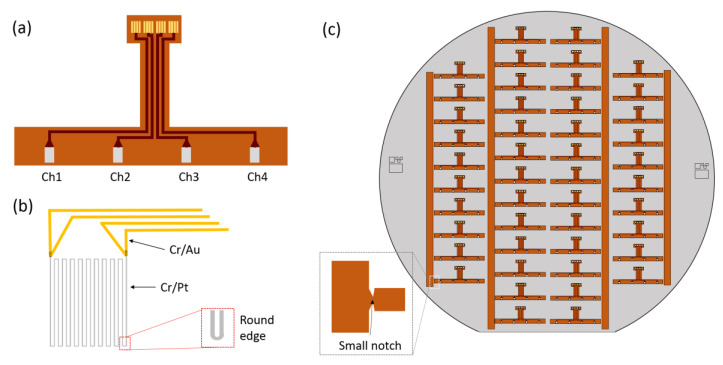
Two-dimensional schematic design of the implantable temperature sensor. (**a**) A 4-channel resistive temperature sensor based on flexible polyimide. (**b**) A single Pt resistive element design with Au interconnection lines. (**c**) Sensor arrays fixed on supporting structures with small notch patterns on a 100 mm silicon wafer.

**Figure 2 sensors-20-03953-f002:**
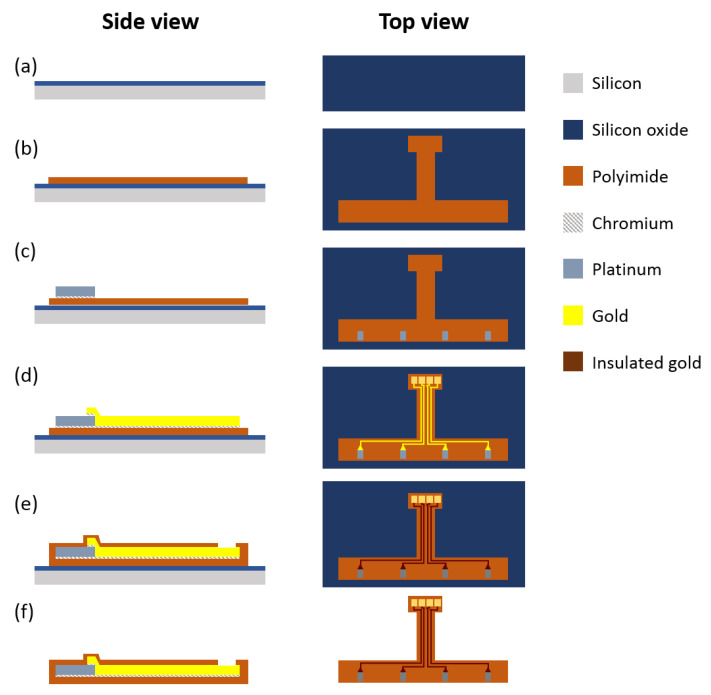
Illustration of the microfabrication process. (**a**) Deposition of a silicon oxide sacrificial layer on a 100 mm bare silicon wafer. (**b**) Deposition of the first polyimide layer. Micropatterning of (**c**) the Cr/Pt layer for temperature sensor and (**d**) the Cr/Au layer for both interconnection lines and pads. (**e**) Deposition of a second polyimide layer as an insulation layer, maintaining contact openings by photo-lithography. (**f**) Remove the sensor from the silicon wafer.

**Figure 3 sensors-20-03953-f003:**
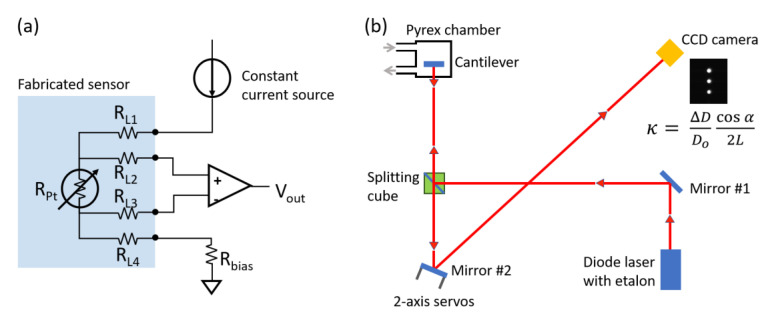
Schematics of (**a**) Four-wire temperature measurement circuit with Kelvin connection and (**b**) in situ cantilever curvature measurement system comprised of a multi-beam optical stress sensor. A typical CCD image is shown. The curvature of the cantilever is determined by the average spot separation.

**Figure 4 sensors-20-03953-f004:**
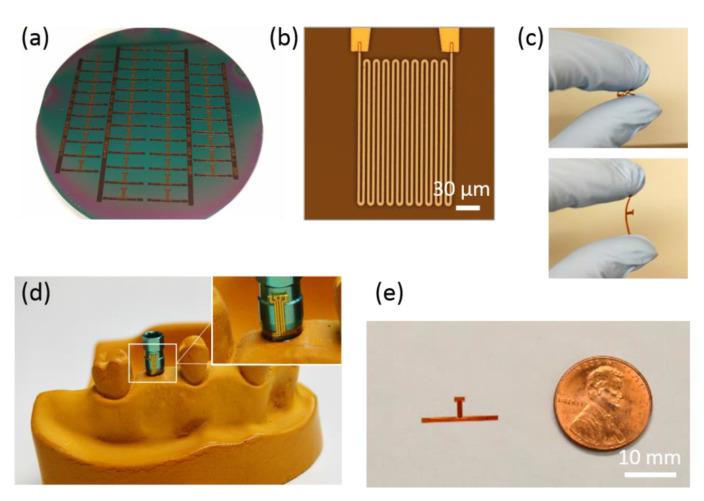
Microfabricated temperature sensors. (**a**) A total of 46 sensors on a silicon wafer before releasing. (**b**) Bright-field microscopic image of one channel of Pt resistor and overlapped Au interconnection lines before covered by top insulation layer. (**c**) Highly flexible sensor without mechanical damage. (**d**) Adhered temperature sensor around an abutment wing of the dental implant platform. (**e**) Developed temperature sensor smaller than a penny.

**Figure 5 sensors-20-03953-f005:**
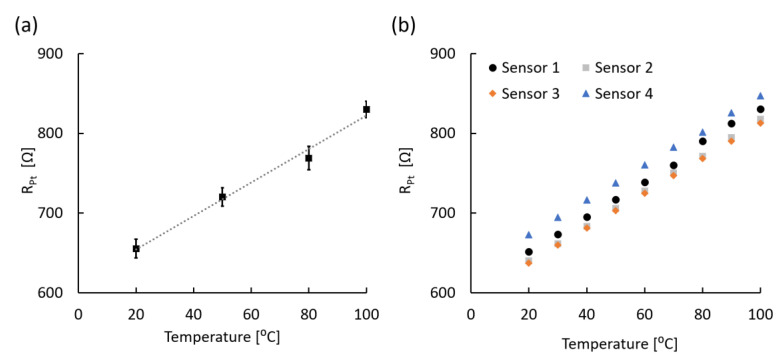
Linear resistance change of developed sensors depending on temperature change between 20 and 100 °C. (**a**) Averaged resistance plot with linear regression (dotted line; R^2^ > 98%; N = 20). Error bars represent the standard deviation (SD). (**b**) Four examples of resistance of independent sensors from multiple batches.

**Figure 6 sensors-20-03953-f006:**
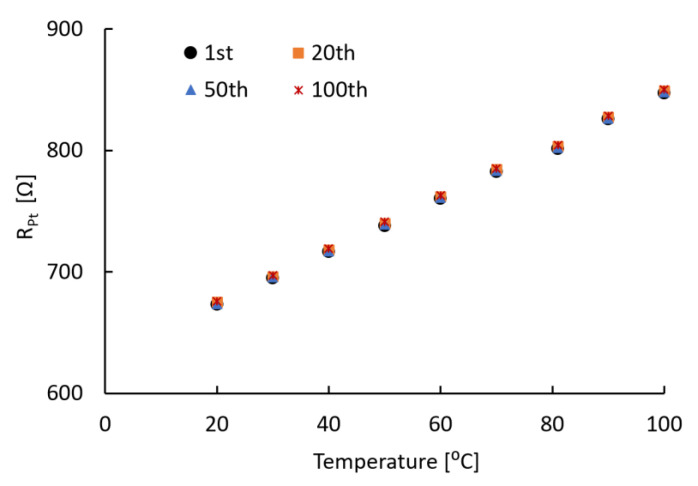
Resistance of Pt resistor for 100 sweeps of the temperature between 20 and 100 °C.

**Figure 7 sensors-20-03953-f007:**
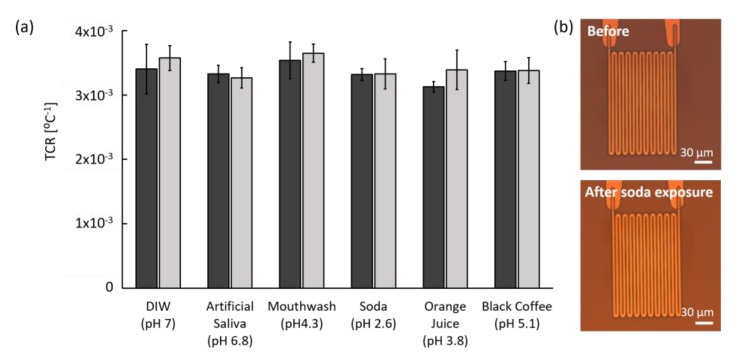
Chemical stability results before (dark gray) and after (light gray) chemical exposure. (**a**) Stable TCR values before and after immersion in DI water, artificial saliva, and four acidic solutions for 6 to 7 d (N = 19). Error bars represent the SD. (**b**) Example of microscopic polarized images of the sensor surface before and after exposure to soda for 7 d.

**Figure 8 sensors-20-03953-f008:**
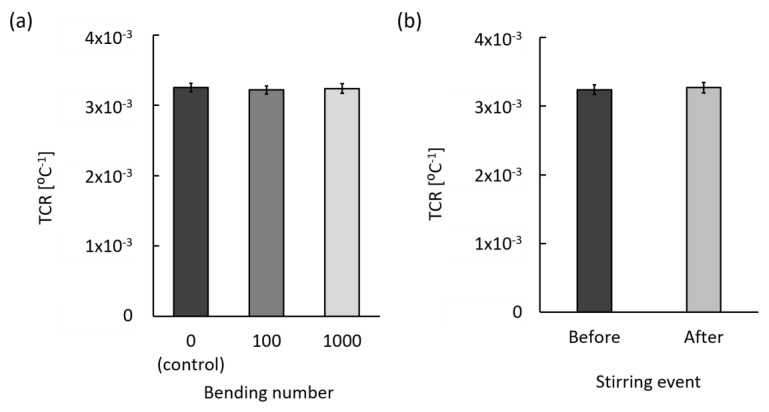
Stable mechanical properties before and after the mechanical stress test: (**a**) 180° bending test for 1000 times (N = 9) and (**b**) stirring test for 30 min (N = 6). Error bars represent the SD.

**Figure 9 sensors-20-03953-f009:**
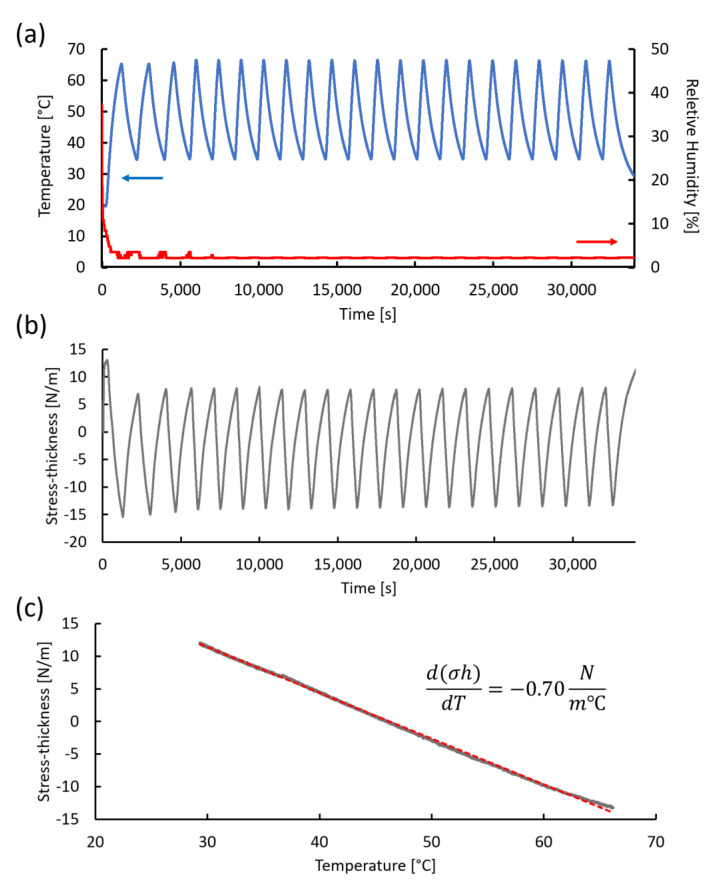
Thermal stress measurement using in situ cantilever curvature. (**a**) Change in temperature (blue) and relative humidity (red) during a typical thermal cycling experiment. (**b**) Stress-thickness response during 22 thermal cycles. (**c**) A plot of stress-thickness vs. temperature during the final cooling cycle. The red dotted line represents a linear regression line with a slope of −0.70 N/(m°C).

**Figure 10 sensors-20-03953-f010:**
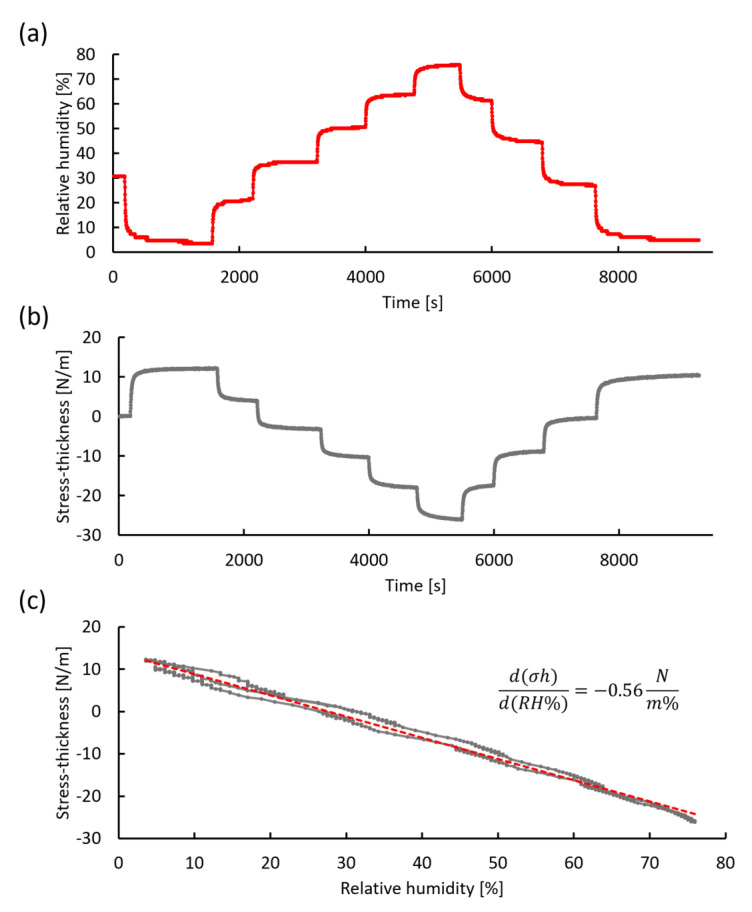
Humidity stress measurement using in situ cantilever curvature. (**a**) Change in relative humidity produced by mixing dry air and humidified air with a total flow rate of (500 ± 5) mL/min. (**b**) Stress-thickness response during humidity changes. (**c**) A plot (gray dots) of stress-thickness vs. relative humidity shown in (a) and (b). The red dotted line represents a linear regression line with a slope of −0.56 N/(m%).

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
