# Peer review of "Development of a Dental Implantable Temperature Sensor for Real-Time Diagnosis of Infectious Disease"

_sensors, 2020, doi:10.3390/s20143953_

Round 1

Reviewer 1 Report

The authors present a publication in which they describe the development of a multichannel temperature sensor for temperature measuring of a dental implant. The article describes in detail the development and construction of the sensor, including its properties. The second part describes in detail verification of the stability of the sensor from various perspectives, which are very important for using in clinical practice.

The structure of the article is good and clear. The used methods have been sufficiently described, the results are presented appropriately and are sufficiently discussed in the end of the article.

New interesting applications and future works can be envisioned.

Author Response

Response to Reviewer 1 Comments

We appreciate Reviewer 1 for the careful review and constructive suggestions. Below is our response to Reviewer 1’s comment on future application.

Comment: New interesting applications and future works can be envisioned.

Response: In response to this comment, we added a new paragraph in the Discussion section to describe our future work and new applications.

Revised manuscript (Page 16, line 503-508):

“Therefore, we plan to further upgrade the sensor by further developing the microfabrication process and testing implantable capacity. First, we plan to integrate wireless communication and battery-recharging capabilities to the dental implant platform. Also, with further microfabrication development, we plan to add physio-chemical sensing mechanism to the temperature sensor for simultaneous monitoring of multiple parameters. Lastly, we anticipate upgrading our sensor to a sensor-embedded chair-side dental instrument that can then be tested for oral disease diagnosis.”

Reviewer 2 Report

Sensors Manuscript Number: sensors-847948 Title: "Development of Dental Implantable Temperature Sensor for Real-time Diagnosis of Infectious Disease" Author(s): Jeffrey J. Kim, Gery R. Stafford, Carlos Beauchamp and Shin Ae Kim   The work describes the microfabrication of an implantable temperature sensor based on noble metals micropatterned on polyimide flexible substrates. The authors describe detailed steps of the microfabrication process of the sensor and they realize the setup of the temperature measurements circuit. Then, the attention is drawn to the external polyimide substrate of the sensor, thus, its physical, chemical, mechanical and thermal stability have been tested.   The current form of the paper is well written and the characterization techniques are logically chosen, the scientific subject is of great importance and it is related to the biomedical engineering sector, from practical point of view.   Yet, the main CONCERN here is related to the safety of the chosen polyimide. Of course, it is said that this type of polymers present biocompatibility, but, none of the two cited papers (ref 24 & 25) do not contain studies performed on the polyimide used in this microfabricated sensor. So, it is expected that the biocompatibility of the polyimide used in the current study to be verified. Some scientific references dealing with such studies on exactly this HD-8820 polyimide, will be also acknowledged.   If not, the authors must think to check the biocompatibility of HD-8820 polyimide BEFORE the current work to be qualified for publishing in Sensors.

Reviewer 3 Report

The paper entitled "Development of Dental Implantable Temperature
3 Sensor for Real-time Diagnosis of Infectious Disease.” concerning use of a dental implantable temperature sensor that can send early warning signals in real time before a dental implant fails.

I have read with a great interest the paper. As implantologist who treat peri-implantitis I can't find practical benefit of this sensor when treating-peri implantitis.
The gold standard in peri-implantitis diagnosis is visual assessment of soft tissues, probing and roentgenographic analysis. In my opinion this sensor seems to have no additional benefit in peri-implantitis diagnosis.
For this reson I do not reccomend to publish the paper in the Sensors journal.

Reviewer 4 Report

The paper presents a device to measure the temperature in "real-time" in a dental implant in "log-term".

A few questions mainly on the motivation and terms use to describe the paper:

1- temperature measurement is inherently "real-time". I argue you cannot measure the temperature "off-line". Real-time usually refers to a system that has "predictable delay". It would be an appropriate term to use if the author referring to temperature measurement during the implantation process, but then I do not understand why the "long-term" part of this sensor is appropriate. I think this motivation needs more clarification. 

2- If the author wants to use the term "continuous", not "real-time" the external current source and peripheral need to the measurement would be the issue 

3- If the application is for dental measurement the 20-100 degree temperature seems to be very non-physiological. 

4- It is not clear how much accuracy and precision is the need for the application and at what temperature.

5- Please address safety issues especially with the current base measurement.

Round 2

Reviewer 3 Report

I reconsidered my previous decision after the author's explanations. The paper is prepared well with the scientific soundness. The paper can be accepted in the present form.